# Protocol for a randomised controlled trial investigating an intervention to boost decentering in response to distressing mental experiences during adolescence: the decentering in adolescence study (DECADES)

Marc P Bennett [ID],[1] Rachel Clare Knight [ID],[1] Darren Dunning [ID],[1] Alan Archer-Boyd,[1] Sarah-Jayne Blakemore [ID],[2,3] Edwin Dalmaijer [ID],[1] Tamsin Ford [ID],[4] J Mark G Williams [ID],[5] Hannah Clegg,[1] Willem Kuyken [ID],[5] Tierney So,[6] Gemma Wright,[1] Bert Lenaert,[7] Maris Vainre [ID],[1] Peter Watson [ID],[1] MYRIAD Team, Tim Dalgleish [ID] [1]

MPB and RCK are joint first authors.

For numbered affiliations see end of article.

**Correspondence to**
Miss Rachel Clare Knight;
rachel.knight@mrc-cbu.cam.ac.uk

## ABSTRACT

**Introduction** Decentering describes the ability to voluntarily adopt an objective self-perspective from which to notice internal, typically distressing, stressors (eg, difficult thoughts, memories and feelings). The reinforcement of this skill may be an active ingredient through which different psychological interventions accrue reductions in anxiety and/or depression. However, it is unclear if decentering can be selectively trained at a young age and if this might reduce psychological distress. The aim of the current trial is to address this research gap.

**Methods and analysis** Adolescents, recruited from schools in the UK and Ireland (n=57 per group, age range=16–19 years), will be randomised to complete 5 weeks of decentering training, or an active control group that will take part in a combination of light physical exercise and cognitive training. The coprimary training outcomes include a self-reported decentering inventory (ie, the Experiences Questionnaire) and the momentary use of decentering in response to psychological stressors, using experience sampling. The secondary mental health outcomes will include self-reported inventories of depression and anxiety symptoms, as well as psychological well-being. Initial statistical analysis will use between-group analysis of covariance to estimate the effect of training condition on self-rated inventories, adjusted for baseline scores. Additionally, experience sampling data will be examined using hierarchical linear models.

**Ethics and dissemination** This study was approved by the Cambridge Psychology Research Ethics Committee, University of Cambridge (PRE.2019.109). Findings will be disseminated through typical academic routes including poster/paper presentations at (inter)national conferences, academic institutes and through publication in peer-reviewed journals.

**Trial registration number** ISRCTN14329613.

## Strengths and limitations of this study

► We aim to strengthen an active and pan-therapeutic skill that features across different psychological interventions, and may improve well-being in adolescents at high risk of mental health difficulty.
► The intervention will be highly accessible by capitalising on extant social media and streaming services.
► Ecological momentary assessment will facilitate a temporally sensitive investigation into the effects of decentering training in everyday life.
► A lack of face-to-face interaction between experimenters and participants may undermine adherence.
► The study may be demanding as it requires daily, although brief, engagement over a several weeks.

## INTRODUCTION
### Background
Anxiety and depression are a considerable public health challenge. Since these difficulties often begin before the age of 24, responding to this challenge requires a special focus on adolescence.[1] It has been suggested that an effective way to promote early mental health is through universal approaches[2–4]; these are interventions that can be offered across a broad community of young people, irrespective of symptom severity. The goal of such universal approaches is to not only reduce distress in young people experiencing symptoms, but also to prevent future anxiety and depression onset in those who may be asymptomatic. However, their effectiveness depends on how well an intervention focuses on psychological experiences

and mechanisms that are relevant across the entire spectrum of mental health, from risk to resilience and through to flourishing. One strategy is therefore to target widely relevant therapeutic skills that can help manage distressing inner experiences (eg, unpleasant feelings, thoughts, memories), which can otherwise trigger short-term distress and long-term risk of anxiety and depression onset. Psychological decentering, or decentering for short, may be one such skill.

Psychological decentering is a key concept within psychological therapy and science.[5] It is characterised as an adaptive self-observation style wherein one can attend to, not just the mental content of distressing inner experiences, but also its underlying cognitive nature.[6 7] Thus, decentering relates to how a person interacts with the difficult inner events prompted by experiences in daily life—it involves the ability to generate an objective self-perspective wherein one can notice mental experiences as imperfect models of the real-world rather than precise reflections. For example, a person might notice 'I am thinking I am depressed right now' instead of only noticing and then believing the thought 'I am depressed'.[8] Decentering resultantly restricts the disproportionate influence psychological stressors may have on affect, behaviour and sense-of-self. Indeed, brief therapeutic exercises that stimulate a decentred self-perspective have been shown to reduce negative emotional reactivity towards stressors like unpleasant memories and negative self-relevant statements.[9–13] Evidence also indicates self-reported decentering traits in adolescence are associated with fewer overall symptoms of anxiety and depression.[14–17]

Decentering is a malleable skill that can be refined through psychological intervention.[7] Medium-to-large changes in self-reported decentering are reported in adults experiencing anxiety and depression following a range of psychological interventions. This includes cognitive behavioural therapy,[18 19] acceptance-based approaches[20–22] and mindfulness training.[23–25] In fact, emerging evidence suggests that intervention-related increases in decentering are a common pathway through which different psychological interventions deliver reductions in anxiety and depression.[21 26 27] This has encouraged some investigations into whether or not decentering can be trained selectively so to improve mental health. For example, Travers-Hill *et al* trained adults with a diagnosis of recurrent depression currently in remission but experiencing residual depression symptoms to practice self-distancing techniques in response to difficult autobiographical memories.[28] This resulted in a significant increase in self-reported measures of decentering relative to an active comparison group who learnt strategies to manage maladaptive avoidance tendencies. Additionally, those who learnt to apply this decentering-related technique reported less distress towards negative memories at post intervention even in the absence of instructions to self-distance.

Therapeutic exercises that target decentering skills are already nested within different psychological interventions.[7] Emerging evidence also indicates that these components can be delivered in isolation so as to selectively boost decentering and reduce risk of relapse in formerly depressed adults.[28] However, a remaining question is 'can decentering skills be selectively reinforced during adolescence, and if so, through what therapeutic techniques?' This is the primary research question guiding the current study. It is also unclear if boosting adolescent decentering skills might impact on emotion and mental health. The current study will therefore evaluate, as a secondary question, how decentering training impacts anxiety, depression and emotional reactivity in adolescence. This information will inform future research trials and ultimately help in the investigation of effective universal approaches in youth mental health. A straightforward way to address these questions might be to examine the impact of decentering training on trait-level measures of decentering and symptom severity at critical time points, for example, pre and postintervention design. However, this approach lacks the sensitivity required to detect subtle changes in decentering in response to momentary psychological stressors and their impact on emotion. A design that also includes ecological momentary assessment or experience sampling methods (ESM) could therefore facilitate a more ecological investigation of the use of decentering in everyday life as well as its impact on transient psychological stressors and emotional states.

## Current study objectives

The primary aim of the current study is to investigate if decentering can be improved during adolescence via an intensive decentering training programme. This will be investigated (1a) at a trait level by examining changes in self-reported experience of decentering (via a standard decentering inventory) and (1b) at a situational level by examining self-reported use of decentering skills in response to momentary psychological stressors (via ESM assessments). Here, repeated measures of participants' negative emotional reactivity towards momentary inner experiences (eg, feelings, thoughts and memories) will be recorded across the duration of the intervention. A secondary aim is to explore potential mental health outcomes associated with reinforced decentering skills. This will be investigated (2a) at the symptom level by examining relative change in self-rated anxiety, depression and psychological well-being and (2b) at the situational level by recording emotional reactivity towards momentary psychological stressors using ESM. A final exploratory aim is to investigate some of the cognitive correlates of decentering and training. Thus, this study will answer the following research questions: (1a) Is decentering training during adolescence associated with increased decentering reports relative to an active control condition? (1b) Is decentering training during adolescence associated with increased decentering in response to negative mental events relative to an active control condition? (2a) What is the impact of decentering training on youth mental

health relative to an active control condition? (2b) What is the impact of decentering training on emotional reactivity towards momentary negative mental events relative to an active control? (3) What are the cognitive correlates of decentering training during adolescence?

This study will address these research questions in a cohort of adolescents who are at increased risk of depression. This is a first step towards our long-term goal to develop a universal intervention that targets psychological experiences and mechanisms that are broadly relevant in young people. Adolescents experiencing elevated symptoms of depression will be recruited from both the UK (eg, sixth form colleges) and Ireland (eg, secondary level education high schools) and will be randomly assigned to a 5-week decentering training programme or an active control. Both programmes will be delivered remotely via extant social media platforms; this means they can be delivered at low cost, are easily accessible and are flexible regarding other commitments of our participants. The decentering intervention comprises 10–15 min of audio exercises to be made available each weekday through music/podcast streaming services, for example, *Spotify, iTunes, Deezer, Podcast Addict*. Participants will practice a specific decentering technique each week, which has been adapted from either the literature or an extant psychological therapy. Participants will also practice brief mindfulness-based grounding exercises each week to encourage the monitoring of inner experiences like emotional thoughts, memories and feelings. The overall goal of this programme is therefore to teach adolescents different ways to generate an objective self-perspective from which they can interact with negative psychological events encountered in daily life. Rather than rehearsing decentering techniques, the active control condition comprises 10–15 min of gamified cognitive tasks for completion on a personal smartphone. In addition, participants in this condition will be given short physical movement routines to complete in lieu of mindfulness-based grounding.

## METHODS
### Study design
The design is a randomised controlled feasibility trial (comparing a psychological decentering training to an active control programme) with school aged adolescents (aged 16–19 years). The study will run from December 2021 until July 2022, split into several cohorts. Self-rated inventories of training and mental health outcomes will be assessed at three time points: within 1 week prior to the start of programme (baseline); within the third week of training (mid-intervention); and 1 week after the programme has finished (postintervention).

### Participants and inclusion/exclusion criteria
Two groups of older adolescents (n=57 per group; age range=16–19 years) will be recruited from the UK and Ireland. Eligible participants must consent to completing

(1) a 5-day ESM baseline assessment, (2) 5 weeks of the assigned training programme and (3) assessment measures at baseline, mid-intervention and post intervention. In exchange for their time, participants will be compensated with shopping vouchers worth (1) £100 pounds (for completing 5 weeks of training) and (2) £20 (as a bonus for completing all assessments). Eligible participants must also report a score of 16 or above on the Centre for Epidemiological Studies-Depression Scale (CES-D), based on standard CES-D cut-offs. Participants must also have access to a laptop/desktop computer and a personal smartphone device.

Participants will be excluded if they: (1) currently take part in a regular (once or more per week) yoga and/or mindfulness class/workshop, (2) have participated in prior formal meditation training or a mindfulness-based stress reduction course, (3) are currently experiencing chronic illness (eg, epilepsy, chronic pain, cancer), (4) lack fluency in English, (5) have a recent diagnosis of, and are currently receiving medical/psychological treatment for, a mental health condition including (but not limited to) anxiety disorder, major depressive disorder or a traumatic stress disorder and (6) have a diagnosis of a neurodevelopmental condition such as autism spectrum disorder or attention deficit/hyperactivity disorder. Both the decentering programme and active control programme require adolescents to be without severe hearing difficulties since both conditions require listening to audiotapes. As the control condition involves physical movement routines, it may not be appropriate for specific individuals, such as individuals with limited mobility. Any volunteer who meets the exclusion criteria will be accommodated as best as possible (eg, assignment to the most appropriate condition in the case of a physical disability) and their data will not be included in the analysis.

### Recruitment
Participants will be recruited from the general community through a combination of existing collaborations with schools in the UK and Ireland. We will also recruit UK and Irish volunteers via online research platforms (eg, Prolific; www.prolific.co) as well as extant panels of adolescent research volunteers via the MRC-Cognition and Brain Sciences Unit (University of Cambridge). Finally, and if necessary, we will also conduct a targeted recruitment campaign to recruit young people and schools using posters, pamphlets and online adverts on social media.

### Sample size
The primary training outcome is self-rated decentering and this will be assessed using both a popular inventory of decentering (Experiences Questionnaire; EQ[8]) and ESM items. There is no consensus on how to best conduct a statistical power analysis for ESM data. Power and sample calculations were therefore based on the effect of training condition on EQ scores. A power analysis calculated in G*Power indicated that a total sample size of 90 is powered

at 80% to observe a significant effect of training condition (decentering vs control) on postintervention decentering with a medium effect size of f=0.3 (α=0.05) after adjusting for baseline EQ score. A medium effect size is plausible given that previous research reports a medium–large effect of extant psychological intervention on EQ scores. An attrition rate of around 20% is anticipated based on our previous research. Therefore, with 57 participants in each group, the current study is adequately powered to observe a medium to large effect of decentering training on our candidate training outcome measures and to determine the nature of this interaction. Regarding our secondary mental health outcomes (ie, anxiety, depression and psychological well-being), a sample size of 57 per group is consistent with our previous research to evaluate the impact of novel psychological interventions. That is, such group sizes can provide a reasonable range of point estimates of the effect on mental health outcome measures that are sufficient to guide later research; for example, 57 participants per group is powered at 88% to observe a significant (p<0.05) main effect of group (decentering vs active control) on baseline adjusted mental health outcomes with an effect size f=0.26.

## Intervention
### Decentering training
A 5-week psychological decentering training programme was developed by MPB, RCK and TD, based on our previous protocol.[28] This involves audio-recorded scripts and an accompanying work book that guides participants through four types of decentering techniques; this structure is partly based on a recent taxonomy of self-distancing (a construct closely related to decentering).[29] The decentering techniques include: (week 1) *spatial distancing* wherein individuals are taught to reimagine negative memories from a physically distant perspective (eg, 'replay the memory but as if you're a fly on the wall'); (week 2) *verbal distancing/cognitive defusion* wherein individuals are taught to rephrase negative self-relevant statements in a way that challenges its literal value and influence over affective behaviour (eg, replacing first person pronouns with one's name); (week 3) *temporal distancing* wherein individuals are taught to reconsider specific worries from a temporally distant future (eg, 'how would this seem in 5 years?'); and (week 4) *objective distancing* wherein individuals are taught to adopt a third-person perspective towards negative memories (eg, 'what is the effect of reliving a difficult memory from the perspective of an objective observer?). Week 5 is a revision week, during which participants will be encouraged to practice a different technique from weeks 1 to 4 each day. These techniques were selected since they are directly targeted at how adolescents relate to and observe day-to-day psychological stressors. Specifically, the goal is to teach adolescents concrete ways to generate an objective (or distanced) self-perspective in response to everyday feelings, thoughts and memories that are unpleasant.

We assume this training will develop participants' decentering ability above their baseline levels.

Each week will involve 5 audio-recorded exercises (10–15 min) that will be made available using common streaming services. One exercise will be posted each weekday, Monday to Friday. The first two exercises (Monday and Tuesday) are brief mindfulness grounding exercises designed to promote open monitoring of psychological experiences. The next three exercises (Wednesday to Friday) are decentering training exercises as described above. Adherence to the programme will be encouraged by directly contacting the participants prior to week 1 and at week 3 to discuss their experiences so far and allow troubleshooting. Participant engagement will be monitored by reviewing the number of completed workbook exercises at the end of the 5 weeks. Participants will also complete daily diaries. These will include five questions about the intervention and its application, such as 'Did you complete any of the programme exercises today?' or 'During the day, were you able to apply the skills you've learned from the exercises?' (see online supplemental appendix A). A participant's engagement in the trial will be discontinued if they elect to withdraw their participation or if they experience serious physical/ mental health difficulties that necessitate medical or psychological intervention.

### Physical and cognitive exercise
A 5-week active control programme was developed (MPB, RCK and TD). This contains two elements that roughly match the decentering training for time and cognitive engagement. First, guided physical movement routines will be completed in lieu of mindfulness grounding exercises (Monday–Tuesday). These movements are intended to emulate the physiological nature of grounding exercises but without an internal focus. Participants will watch short videos in which one member of the research team (RCK) illustrates a basic series of body stretches. Accompanying audio provides additional direction and this was recorded using the same voices from the decentering training programme. Each video comprises 15 stretches, with each stretch held for 30 s and a 10 s break between stretches. The physical movements were selected for their ease and accessibility. Care was also taken to select movements that are safe and cater to a range of physical abilities.

Second, gamified versions of standard cognitive tasks will be completed in lieu of the decentering training (Wednesday–Friday). These games are intended to emulate the cognitive effort associated with decentering exercises. Each participant will have a link that allows them to select one of three games, which can be completed on a personal smart phone (for game outlines, see https:// osf.io/aw6c5). Games include: (1) a Multi-Target Visual Search Task wherein participants search and respond to specific targets within a broader stimulus array; (2) a Go/NoGo Task wherein a speeded response is made in response to a 'go' signal but inhibited in response to a

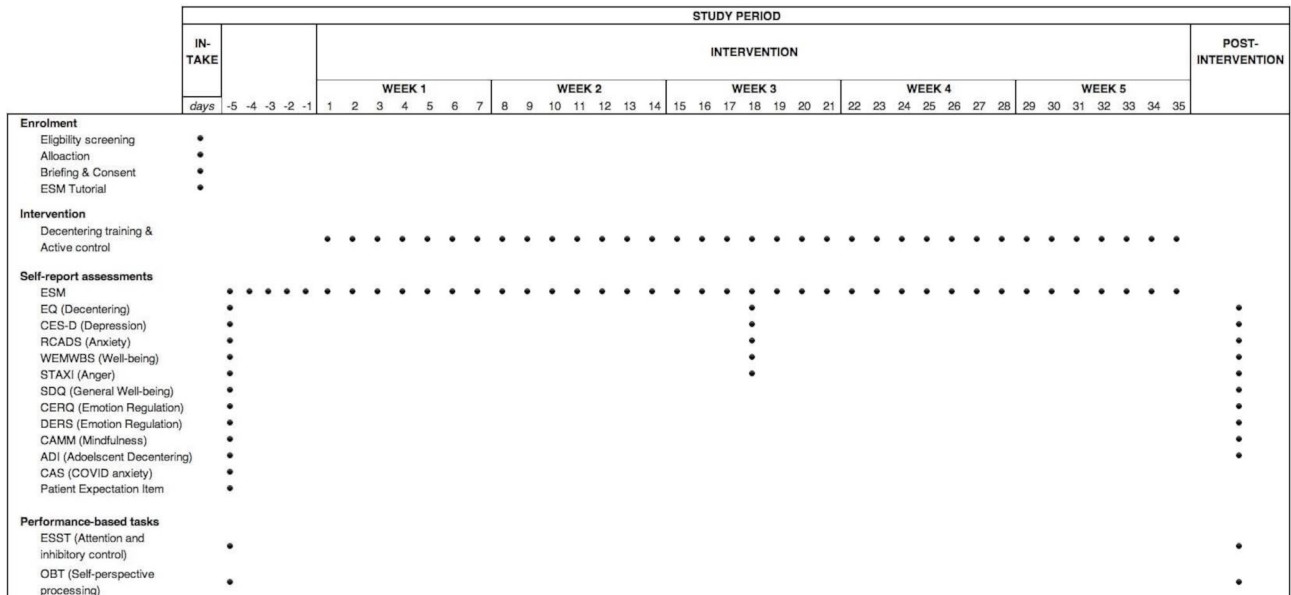

**Figure 1** A schematic overview of the intervention timeline. ADI, Adolescent Decentering Inventory; CAMM, Child and Adolescent Mindfulness Measure; CAS, COVID-19 Anxiety Scale; CERQ, Cognitive Emotion Regulation Questionnaire; CES-D, Center for Epidemiological Studies-Depression Scale; DERS, Difficulties in Emotion Regulation Scale; EQ, Experiences Questionnaire; ESM, experience sampling methods; ESST, Emotional Stop Signal Task; OBT, own-body transformation task; RCADS, Revised Child Anxiety and Depression Scale; SDQ, Strengths and Difficulties Questionnaire; STAXI, State-Trait Anger Expression Inventory-2 Child and Adolescent; WEMWBS, Warwick-Edinburgh Mental Well-being Scale.

'no go' signal; and (3) a Digit Recall Task wherein participants must recall a sequence on a number pad (ie, a digit-span task). Adherence to the programme will be encouraged by contacting the participants prior to week 1 and at week 3 to discuss their experiences. Engagement will be monitored by reviewing task completion rates and performance measures (response times and accuracy) at the end of the 5 weeks.

## Outcomes

There are two types of outcome measures in this study (see figure 1 for timeline). These include measures of (1) decentering and (2) mental health. We will also examine if there are any far-transfer effects of decentering training by including measures of (3) cognitive performance. Figure 1 provides a schematic overview of the assessment timeline.

### Decentering outcome measures

The primary training outcome measure is self-rated decentering as measured using the EQ. A secondary decentering outcome measure is the self-rated use of psychological decentering in response to psychological stressors as measured using ESM. These measures have been included to address research questions 1a and 1b, respectively.

### *Experiences questionnaire*

This widely-used self-report assessment features 11 items that explore an individual's tendency to psychologically decenter from difficult subjective experiences in day-to-day life (eg, 'I can observe unpleasant feelings without being drawn into them').[8] Items are answered using a 5-point

Likert scale from 1='Never' to 5='All the time'). Psychometric properties were found to be satisfactory (Cronbach's α=0.893; convergent validity r>0.46).[30] Scores on the EQ are positively associated with other decentering-related constructs like experiential avoidance (a tendency to attempt to evade difficult thoughts and feelings)[30–34] and cognitive reappraisal (an ability to reconceptualise situations to modify emotional impact).[27 35] Finally, EQ scores improve over the course of behavioural and cognitive therapies; the magnitude of such change is associated with key outcomes like symptom severity and quality of life.[23 26 27 36 37] The EQ will be administered at baseline, mid-intervention and post intervention. We expect mean EQ scores will increase in the decentering training group relative to the active control condition.

### *ESM decentering items*

All ESM items are described in table 1. Some items relate primarily to the use of decentering skills. Other ESM items relate to secondary mental health outcomes and are described in the next section. However, all ESM items will be administered during each sample point. Two ESM items have been developed to estimate the momentary use of decentering in response to psychological stressors like difficult feelings, memories or thoughts. These are 'Since the last beep, I was able to distance myself from unpleasant feelings' and 'Since the last beep, I was able to distance myself from unpleasant things on my mind'. Items are answered using a 7-point Likert scale ranging from 1='Not at all' to 7='Very much'). ESM items will be delivered 4 times daily across a 5-day ESM baseline period and the 5 weeks of training (see figure 1). All ESM

| Item | Question | Rating | Purpose | Conditional |
|------|----------|--------|---------|-------------|
| *1* | Right now, I feel… | −5=Very negative, 5=Very positive | Affect | Always shown |
| *2* | I Feel: | | Feelings/mood | Always shown |
| *i* | *Happy* | 1=Not at all, 7=Very much | Feelings/mood | Always shown |
| *ii* | *Relaxed* | 1=Not at all, 7=Very much | Feelings/mood | Always shown |
| *iii* | *Satisfied* | 1=Not at all, 7=Very much | Feelings/mood | Always shown |
| *iv* | *Enthusiastic* | 1=Not at all, 7=Very much | Feelings/mood | Always shown |
| *v* | *Nervous* | 1=Not at all, 7=Very much | Feelings/mood | Always shown |
| *vi* | *Sad* | 1=Not at all, 7=Very much | Feelings/mood | Always shown |
| *vii* | *Irritated* | 1=Not at all, 7=Very much | Feelings/mood | Always shown |
| *viii* | *Stressed* | 1=Not at all, 7=Very much | Feelings/mood | Always shown |
| *3* | Since the last beep, I felt sucked in by negative feelings? | 1=Not at all, 7=Very much | ER \| Immersion negative feelings | Always shown |
| *4* | Since the last beep, I was able to distance myself from negative feelings? | 1=Not at all, 7=Very much | ER \| Distance negative feelings | Always shown |
| *5* | Since the last beep, I tried to distract myself from negative feelings? | 1=Not at all, 7=Very much | ER \| Distraction | Always shown |
| *6* | Since the last beep, I noticed an unpleasant thought or memory? | 1=Not at all, 7=Very much | Thoughts/memories onset | Always shown |
| *7* | Since the last beep, I was upset by an unpleasant thought or memory? | 1=Not at all, 7=Very much | Thoughts/memories onset | Always shown |
| 8 | The unpleasant thing on my mind was about … | Myself, Others, A combination of these, None of these | Content—self/other | Contingent \| (if Q6 !=1) |
| 9 | The unpleasant thing on my mind was about … | Social things, Professional things, A combination of these, None of these | Content—social/non-social | Contingent \| (if Q6 !=1) |
| 10 | The unpleasant thing on my mind was about … | The past, The present, The future, A combination, None of these | Content—past/future | Contingent \| (if Q6 !=1) |
| *12* | Since the last beep, I felt sucked in by unpleasant things on my mind. | 1=Not at all, 7=Very much | ER \| Immersion in thoughts/memories | Always shown |
| 13 | Since the last beep, I able to distance myself from unpleasant things on my mind. | 1=Not at all, 7=Very much | ER \| Distance from thoughts/memories | Always shown |
| *14* | Since the last beep, I tried to think differently about things so to feel better. | 1=Not at all, 7=Very much | ER \| Reappraisal | Always shown |
| 15 | Since the last beep, I tried to distract myself from unpleasant things in my mind. | 1=Not at all, 7=Very much | ER \| Distract | Always shown |
| *16* | Since the last beep, I feel I benefited from the daily programme. | 1=Not at all, 7=Very much | Usefulness | Always shown |
| *Branch 1* | Right now, I am … | Indoors, Outdoors | Branch | Contingent \| (if Q6==1) |
| *Branch 2* | Right now, I am … | Alone, With others | Branch | Contingent \| (if Q6==1) |
| *Branch 3* | Since the last beep, I felt social. | 1=Not at all, 7=Very much | Branch | Contingent \| (if Q6==1) |

**Table 1** Experience sampling items

ER = Emotion Regulation

items are described in table 1 (see overleaf). We expect momentary reports of decentering to increase over time in the decentering training group relative to the active control condition.

### Secondary mental health outcomes

Our secondary mental health outcomes are (1) self-rated anxiety, depression and psychological well-being and (2) self-rated emotional reactivity following a negative mental experience (eg, unpleasant thought, feeling or memory)

as measured using ESM. These measures have been included to address research questions 2a and 2b.

### Mental health symptoms

This study will explore the potential impact of decentering training on mental health outcomes at baseline, mid-intervention and post intervention. Previous findings suggest that decentering is negatively associated with anxiety and depression[14–16] as well as feelings of anger in adolescents.[38 39] We will therefore include measures

of each of these difficulties. Anxiety symptoms will be assessed using the Revised Child Anxiety and Depression Scale-Short Version (RCADS-15).[40] The RCADS-SV is a 15-item scale measuring the reported frequency of various symptoms of anxiety and low mood. Internal consistency has been reported as good (Cronbach's α=0.7-.96), as were test–retest coefficients and convergent validity. Depression symptoms will be assessed using the CES-D.[41] The CES-D is a 20-item inventory measuring depressive symptoms experienced in the past week. Internal consistency is good, with Cronbach's α=0.85–0.90. Concurrent and construct validity have been demonstrated. Anger will be measured using the State-Trait Anger Expression Inventory-2 Child and Adolescent (STAXI-2),[42] which is a 35-item self-report measuring anger expression and control in adolescents. The STAXI exhibits good reliability, with Cronbach's alpha coefficients ranging from 0.81 to 0.93, with good convergent validity.

Two measures will also be included to estimate the impact of decentering training on psychosocial strengths/difficulties and general well-being. The Strengths and Difficulties Questionnaire (SDQ) is a screening tool with five subscales (Emotional Symptoms, Conduct Problems, Hyperactivity/Inattention, Peer Relationships and Prosocial Behaviour) as well as an impact supplement to estimate psychosocial functioning. This tool was included to capture behavioural symptoms such as conduct difficulties and hyperactivity. The SDQ exhibits strong internal consistency, moderate test–retest reliability and good concurrent validity. Psychological well-being will be measured using the Warwick-Edinburgh Mental Well-being Scale (WEMWBS).[43] The WEMWBS is a 14-item measure of positive mental health including positive affect, functioning and interpersonal relationships. The WEMWBS shows high-internal consistency, and good convergent and construct validity. Measures will be administered at baseline, mid-intervention and post intervention.

### ESM emotional reactivity
This study will examine if the lagged effect of the decentering from difficult thoughts, feelings or memories (time=$n$) on later affect (time=$n + 1$) changes across training. One ESM items will capture general affect: 'Right now I feel…', answered on a 11-point Likert scale ranging from −5=Very negative to +5=Very positive. Two separate ESM items will capture the self-initiated use of decentering skills in response to negative mental experiences (table 1). These are: 'Since the last message, I was able to distance myself from [a] a negative feeling *or* [b] unpleasant things on my mind'. Additional ESM items will check for the occurrence of negative mental experiences. These include checks of (a) negative feelings by asking 'I feel Nervous/Sad/Irritated/Stressed' and (b) unpleasant cognitive events by asking 'Since the last beep, I was upset by a thought or memory'. This will be scored on a 7-point Likert scale from 1=Not at all to 7=Very much.

One item will assess the occurrence of unpleasant thoughts: 'since the last beep, I noticed an unpleasant thought or memory'. This will allow us to control for the presence of unpleasant cognitive events from which participants can decenter. If the participant's response is anything other than not at all, they will then complete three items to the content of these thought (table 1, items 8–10). If the participant's response is not at all, they will complete three unrelated 'branching' items (table 1, branch items 1–3). This means the same number of item will always be administered.

### Additional self-rated assessments
Participants will complete several additional self-rated assessments for the purpose of exploratory analysis. Participants will also complete a brief end of day questionnaire that will include a measure of mood, assessing presence and duration of different mood states. Nine moods will be investigated—happy, lively, content, satisfied, depressed, bored, anxious, irritable and tense. These items will allow for exploratory analysis of how individuals' moods may relate to the two experimental conditions. The end of day questionnaire will also include items investigating whether participants have completed any exercises from their programme (decentering or control), how difficult these exercises were to complete, and whether these skills were relevant to their everyday life. This section is included both as a measure of adherence to the programme, and as a broad measure of participant feedback.

Our team is currently developing an adolescent friendly self-rated decentering inventory. A provisional version of this scale will be included at pre and post intervention as part of its on-going validation. We are also investigating the relationship between decentering and other emotion regulation skills. We will therefore include: the Cognitive Emotion Regulation Questionnaire,[44] which is a 36-item questionnaire cataloguing a range of emotion regulation strategies (Cronbach's α=0.62–.85); the Difficulties in Emotion Regulation Scale,[45] which is a 36-item questionnaire investigating problems with emotion regulation (Cronbach's α=0.94), including six subscales: non-acceptance of emotional responses; difficulty in engaging in goal-directed behaviour; impulse control difficulties; lack of emotional awareness; limited access to emotion regulation strategies; and lack of emotional clarity: and the Child and Adolescent Mindfulness Measure,[46] which is a 10-item measure exploring trait mindfulness in children and adolescents (Cronbach's α=0.88). Participants will complete the Coronavirus Anxiety Scale[47] scales at baseline and post-intervention. This may help us address any unexpected impact of the COVID-19 pandemic on the intervention.

### Far transfer effects
This study will include performance-based tasks to explore potential far-transfer effects of decentering training on cognition (research question 3). These include (1) affective cognitive control and (2) self-perspective processing,

both of which have been posited as theoretical components of decentering.[6 29] These measures will be administered at baseline and post intervention.

### Affective cognitive control

This study will explore if decentering training influences the ability to maintain cognitive performance within emotional contexts (ie, affective cognitive control). This can be estimated via the Emotional Stop Signal Task (online supplemental figure S1).[48] On each trial, a neutral or negative valence image is presented before a 'go-signal' that requires a speeded button-press (75% of trials). Participants must quickly make the appropriate button-press unless the go-signal is followed by a 'stop-signal' (25%). Participants must withhold their response on these trials. The period between the go and stop-signal (stop signal delay; SSD) varies over the task. Following a successful stop trial, the task is made more difficult by increasing the SSD by 25 ms. Following an unsuccessful stop trial, the task is made easier by decreasing the SSD by 25 m. This tracking algorithm typically causes a failure to stop on ~50% of trials. Sustained attention in emotional and non-emotional contexts will be estimated by calculating (1) the reaction time (RT) mean on go-trials and (2) the intertrial variability coefficient (ie, the RT mean divided by the RT SD). Inhibitory control in emotional and non-emotional contexts will estimated by calculating the *stop signal reaction time (*ie, mean RT–mean SSD).

### Self-referential processing

This study will explore if decentering training influences the ability to flexibly shift one's self-perspective. This can be indexed using the own-body transformation task (online supplemental figure S2).[49] On each trial, a schematic human figure is presented that face either with their front to the participant or their back to the participant. This figure also holds a black glove in one hand (eg, left). Participants are instructed to imagine themselves in the body position of the figure and then judge whether the glove is held by the figures left of right hand. Performance is compared with a within-subject control task wherein participants judge the position of the glove without altering their self-representation. The outcome variables are mean RT and the percentage error rate. These are found to increase as a function of the self-perspective manipulation; that is, they are higher for front-facing characters.

## Procedure

### Data quality

Checks of data quality will be included ad different stages of data collection and analysis. First, 'catch questions' will be embedded in self-report assessments to assess whether participants actively read questions (eg, 'Please select "Not at all"'). Second, participants self-rate the quality of their data at the end of each data collection period, for example, []. Piloting revealed this to be a useful strategy. Third, all the standard markers of appropriate

behavioural performance will be checked when analysing task data (eg, RTs>250 ms and <3000 ms; chance-level accuracy).

## Assignment of interventions

### Allocation

Following baseline assessment, participants will be stratified according to sex and depression severity (using the CES-D). They will then be randomly assigned to either the decentering training programme or the active control condition. This will be managed by the trial statistician (PW) using a minimisation procedure. Participant allocation will be then shared with the research coordinator responsible for providing appropriate access links for the online intervention (MPB, RCK and HC).

### Blinding

Research coordinators (MPB and RCK) will be blinded during assessment, data preprocessing and statistical analysis. Due to the nature of the study, it is not possible to remain blinded during participant allocation. This is because each participant needs to be given a secure link to the online training programme, either decentering training or the active control. All outcome assessments will be completed online; participants will access the same link to complete online assessments, regardless of group. Thus, there are no in-person data collectors with an awareness of participants' allocation. Initial data will be preprocessed (eg, calculation of summary scores like means, percentage accuracy, sums, etc). Preprocessing will be completed using pre-existing scripts coded in R and MATLAB (MPB). At this point, the dataset will be shared with the trial statistician (PW) who will conduct the statistical analysis and who will be blind to participant allocation. Once the analysis is complete, the trial coordinators will unblind the dataset by indicating which participants were allocated to which group.

## Consent and data collection

Recruited volunteers will complete an eligibility check during an initial phone call. Volunteers who are eligible will be invited to participate in the study. Written informed consent will be sought after an initial briefing that explains the data collection protocol (see online supplemental appendix B for consent form). This briefing will also include a tutorial on downloading and using the experience sampling app (PsyMate2; www.psymate.eu). Participants will also have the opportunity to ask any questions. To discourage possible contamination, participants will be asked to keep the content of the study confidential from others. Once consent is provided, participants will be given an anonymous trial identification code and added to an encrypted list for allocation. Participants will also receive a URL link to complete baseline measures. Figure 1 illustrates the full schedule for

outcome measures. Participants will receive a tutorial on accessing the intervention materials.

Over 5 weeks, participants in both interventions will access and rehearse exercises (Monday–Friday). Decentering training exercises are based on audio files to be delivered via their preferred music/post-streaming services (eg, Spotify/Podcast Addict). The active control exercises are based on YouTube videos posted on a Private Channel. Each day, the PsyMate2 App will prompt participants to complete their daily exercise as well as complete any required ESM items. These will be sent four times per day. A brief survey will also be sent through email each evening to ask whether participants completed their daily exercises, and if so, to ask about their experiences as well as questions on their mood states. At the end of week 5, participants will be asked to complete the online assessment battery. Participants will then complete a debriefing session. During the debriefing session, they will be given the opportunity to provide feedback on their experience of the study, and to ask any questions they may have. They will also be given a list of relevant mental health resources should they feel the need to access them.

### Participant retention and follow-up

Retention will be promoted in a number of ways. First, participants will not be paid until receipt of their completed postintervention assessment battery. Second, participants will be made aware that they can receive a bonus payment of £20 if they complete all of their assessments and daily diaries. Third, telephone contact will be made with participants prior to week 1 and during week 3 to allow any troubleshooting and ensure that participants are filling in diaries and experience sampling appropriately. Telephone contact will also be made with a participant should they fail to complete mid-intervention assessments (week 3).

### Data management and statistical analysis
#### Data confidentiality and management

All participants will be given an anonymous trial identification code to use when submitting their completed assessments. A file linking the participant's name to their identification code will be saved in an encrypted and password protected file and stored on a secure, protected server within the MRC-Cognition and Brain Sciences Unit. This file will be deleted once the final round of data collection is complete. Outcome data will be gathered remotely using online platforms (eg, Qualtrics; www.qualtrics.com) and an ESM data collection app. During periods of data collection, these data will be transferred to in-house secure servers within the MRC-Cognition and Brain Sciences Unit, University of Cambridge. The source data will be then deleted. Access to these data will be restricted to primary research team members (MPB, RCK, TD). Data management will be overseen by these

team members. There will be no formal Data Management Committee because it is a small scale trial.

### Statistical analysis
#### Self-rated and performance-based measures

The statistical analysis will be conducted by the trial statistician (PW) who will be blind to training condition. The analysis will be conducted on an intention-to-treat basis and a similar initial analysis is planned for (1) the primary training outcome (ie, self-rated EQ scores) and (2) secondary mental health outcomes (ie, CES-D, RCADS, STAXI-2, SDQ and WEMWBS). Analysis of covariance will be calculated to estimate the effect of training condition on outcome measures after adjusting for baseline performance and grouping stratification variables. This model will be calculated for our primary end-point, that is, post intervention (ie, postintervention outcome adjusted for baseline score). This model will additionally be calculated for our mid-intervention time point (ie, mid-intervention outcome adjusted for baseline score) and post intervention. There are no planned interim analyses. If a subset of items of a measure (<20% within-measure missing data), a total score will be calculated using the mean score across the non-missing items; an approach we have reported elsewhere (www.osf.io/d6y9q). Otherwise, the score for that measure will be noted as missing. We will assume that data will be missing at random and address this using standard imputation approaches in R (R Core Team, 2013).

#### ESM analysis

Only participants who completed at least 33% of the ESM questionnaires will be included in the analyses, in line with ESM guidelines.[50] Statistical analyses will be performed using hierarchical linear models. These models are especially suited to deal with dependency inherent in the data due their hierarchical structure: ESM observations (level 1) nested within individuals (level 2). Potential confounders will be included in the model as covariates. First, a linear growth model will be estimated to assess the effect of condition (decentering training vs active control) on decentering skills over time (ie, the duration of the intervention). Second, time-lagged analyses will be performed to evaluate if higher self-rated decentering skills when confronted with a momentary psychological stressor (time=n) predict significantly lower negative affect ratings at a later time point (time=n+1). We will also control for momentary negative affect ratings (at time=n). Random intercepts and random slopes models will be estimated. In hierarchical linear models, the fixed effects reflect the overall association between a predictor and the outcome of interest, whereas the random effects reflect individual differences in this association.

### Ethics and dissemination
#### Ethical approval and protocol amendments

This study was approved by the Cambridge Psychology Research Ethics Committee, University of Cambridge

(PRE.2019.109). Approval for any protocol amendments will be sought by this committee. This includes methodological and/or trial management amendments. The online registration of the trial will also be updated in this situation and any amendments will be outlined in the trial manuscripts. This committee will also be contacted should we encounter any unexpected ethical concerns.

## Dissemination

The findings of this study will be disseminated through typical academic routes including poster/paper presentations at (inter)national conferences, academic institutes and through publication in peer-reviewed journals. Given that some research questions are less exploratory than others, two key manuscripts are planned for the trial data. The first will describe the impact of training on self-rated decentering and mental health outcomes. The second will describe the exploratory research around the impact on cognitive performance measures. Findings will also be disseminated to the broader public through seminars and workshops with relevant stakeholders as well as online blogs and podcasts. These manuscripts will be completed and submitted for review irrespective of the nature of the observed effects. Following the publication of the first manuscript, a fully anonymised dataset will be made available online on open-access databases whose servers are located within the UK or Europe. Data processing scripts and programming codes will also be made available online.

## Patient and public involvement

A youth advisory panel was formed from May to December 2020. This panel advised on the accessibility of self-rated measures of decentering. They also provided qualitative feedback on decentering as an emotional regulation stratagem and the potential for decentering training. This took place across several informal focus groups (see, Bennett et al[7] for more information). During study development, patients and the public were not involved in the choice of outcome measures, or recruitment for the study.

Participants of the study will be given opportunity to give feedback in person or anonymously at debrief, so as to improve the protocol for future trials.

## DISCUSSION AND FUTURE RESEARCH

Decentering is a helpful strategy to notice and interact with unpleasant inner experiences. Evidence suggests that the ability to use this strategy is continuously distributed in the population, with those at the higher end of the continuum reporting less anxiety and depression. Decentering also appears to be a malleable skill that is reinforced across a range of different psychological interventions. Selectively training this skill during adolescence might therefore delimit the negative impact of everyday psychological stressors. This has the potential to improve emotional well-being in the short term and mitigate anxiety and depression symptoms in the long term. The current trial is an initial investigation to establish whether decentering skills can be selectively trained during adolescence and to characterise its impact on mental health outcomes. These findings may not only reveal some of the best ways to teach this skill but may provide estimates of effect sizes that can inform future research.

**Author affiliations**
[1]MRC Cognition and Brain Sciences Unit, Cambridge, UK
[2]Department of Psychology, University of Cambridge, Cambridge, UK
[3]UCL Institute of Cognitive Neuroscience, London, UK
[4]Department of Psychiatry, University of Cambridge, Cambridge, UK
[5]Department of Psychiatry, University of Oxford, Oxford, UK
[6]University of Cambridge, Cambridge, UK
[7]Faculty of Healh, Medicine and Life Sciences, Maastricht University, Maastricht, Limburg, The Netherlands

**Collaborators** The MYRIAD Team: Saz Ahmed, PhD, of University College London, Susan Ball, MSc, of University of Exeter, Nicola Dalrymple, MSc, of University of Oxford, Katie Fletcher, HSD, of University of Oxford, Lucy Foulkes, PhD, of University College London, Poushali Ganguli, MSc, of Kings College London, Cait Griffin, MSc, Kirsty Griffiths, MSc, of University of Cambridge, Konstantina Komninidou, BEd, of University of Oxford, Suzannah Laws, BSc, of University of Oxford, Jovita Leung, MSc, of University College London, Jenna Parker, MSc, of University of East Anglia, Blanca Piera Pi-Sunyer, MSc, of University College London, J. Ashok Sakhardande, BSc Hons, Jem Shackleford, MA, MSc, Kate Tudor, PhD, of University of Oxford, and Brian Wainman, BEng, of Plymouth University. These individuals have worked across the MYRIAD strategic award "Promoting Mental Health and Building Resilience in Adolescence: Investigating Mindfulness and Attentional Control"; they are acknowledged as group authors in this article for their substantial contributions to the project development, in accordance with the MYRIAD Dissemination Protocol.

**Contributors** MPB and RCK: substantial contributions to the conception, design of work, data acquisition, data analysis and interpretation, and manuscript preparation. TD, TF, WK, S-JB, JMGW, GW: substantial contributions to the conception and design of work. AA-B, TS, DD, ED, PW, MV, BL, HC: substantial contributions to data acquisition and analysis.

**Funding** This project is funded by a Wellcome Strategic Award (Wellcome Trust, ref 104908/Z/14/z; awarded to TD, S-JB, TF, WK, MW) and by the UK Medical Research Council (Grant Reference: SUAG/043 G101400). The contribution of MPB was partially supported by a Wellcome Trust Active Ingredients in Mental Heatlh Commission. RCK is funded by an Economic and Social Research Council Doctoral Fellowship (ref SUAI/067).

**Competing interests** None declared.

**Patient and public involvement** Patients and/or the public were involved in the design, or conduct, or reporting, or dissemination plans of this research. Refer to the Methods section for further details.

**Patient consent for publication** Not applicable.

**Provenance and peer review** Not commissioned; externally peer reviewed.

and indication of whether changes were made. See: https://creativecommons.org/licenses/by/4.0/.

## ORCID iDs
Marc P Bennett http://orcid.org/0000-0001-7217-4059
Rachel Clare Knight http://orcid.org/0000-0002-8315-2864
Darren Dunning http://orcid.org/0000-0002-7696-4143
Sarah-Jayne Blakemore http://orcid.org/0000-0002-1690-2805
Edwin Dalmaijer http://orcid.org/0000-0003-3241-0760
Tamsin Ford http://orcid.org/0000-0001-5295-4904
J Mark G Williams http://orcid.org/0000-0002-9884-2614
Willem Kuyken http://orcid.org/0000-0002-8596-5252
Maris Vainre http://orcid.org/0000-0001-9570-3726
Peter Watson http://orcid.org/0000-0002-9436-0693
Tim Dalgleish http://orcid.org/0000-0002-7304-2231

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
