## [Reviewer comments · BMJ Open]

ARTICLE DETAILS

TITLE (PROVISIONAL)	Protocol for a randomized controlled trial investigating an intervention to boost decentering in response to distressing mental experiences during adolescence: The Decentering in Adolescence Study (DECADES)
AUTHORS	Bennett, Marc; Knight, Rachel; Dunning, Darren; Archer-Boyd, Alan; Clegg, Hannah; Blakemore, Sarah-Jayne; Lenaert, Bert; Dalmaijer, Edwin; Ford, Tamsin; Kuyken, Willem; Williams, J Mark; Watson, Peter; So, Tierney; Wright, Gem; Vainre, Maris; Dalgleish, Tim

VERSION 1 – REVIEW

REVIEWER	Valdivia-Salas, Sonsoles University of Zaragoza
REVIEW RETURNED	01-Oct-2021

GENERAL COMMENTS	GENERAL COMMENTS Thank you for inviting me to review this protocol for a randomized controlled trial to assess the effect of decentering on adolescents' emotion reactivity, emotion regulation skills, mental health, and cognitive and metacognitive abilities. This a very welcome study, especially because of the target population: adolescents. As the authors point out in the Introduction, the evidence on the benefits of decentering in mental health is growing and consistent. But this evidence is limited to studies with adults. Considering that mental health problems do not appear abruptly but develop over time, focusing on children and adolescents seems of vital importance for the prevention of more severe pathologies during adulthood. In addition, adolescence is, in itself, a stage of development in which profound changes occur that could affect their mental health. For these reasons, recruiting adolescents as the target population seems like a wise choice, and I congratulate the authors for this. I have no objections to the protocol rationale, design, evaluation, and dissemination, but there are some issues I would like to discuss about. 1. In the Article Summary, Limitation's subsection (p. 2), the authors mention the risk of non-adherence because of a lacking face-to-face contact and, possibly, because of the protocol being highly demanding. In any case, I would say these two aspects will inform of how much face-to-face contact and demand is necessary for adolescents to engage in decentering skill training. However, I missed mention to the possible contamination between experimental and control conditions in the remote case that
--

	participants get to know or meet and talk about the procedures. This could be the case if a number of adolescents were recruited from the same school. Do the authors have a plan to avoid possible contamination? Please state. 2. In the Background section, the authors state: "Evidence also indicates self-reported decentering traits in adolescents are associated with fewer overall symptoms of anxiety and depression" (p. 4, lines 15-17) and cite four studies on the role of psychological in/flexibility on mental health symptoms. Hence, I wonder why a measure of psychological in/flexibility is not included in the assessment battery. This would allow for further investigation on the relation between constructs, or even the role of psychological in/flexibility (with proven effects on adolescents' mental health and wellbeing) on the observed outcomes. 3. The N size was confusing. At different points through the Abstract and Methods section, it reads 48 (p.8, line 6), 57 (Abstract and p.9, line 54), and 54 (p.10, line 15) per group. Please clarify. 4. As stated in the Intervention subsection (p. 10), Week 5 of the decentering training is a revision week. How will this revision be conducted? Any new task or exercise? Something like a narrative summary of the skills trained during the previous weeks? Repetition of the exercises practiced before? More information will be appreciated for the sake of replication. 5. As part of the decentering outcome measures, two ESM decentering items will be presented to the participants (p. 13). These two items assess the self-reported ability to distance from unpleasant feelings and thoughts. These are items 4 and 13 of a total 16 included in Table 1. I am a little confused with this ESM: (1) I am assuming items 4 and 13 are not presented alone, but in combination with the whole ESM, is this right? (2) Table 1 lists a total of 16 items, most of them will be presented always, only a few will be presented contingent upon a particular response. Three items (3, 4, 5) assess decentering skills with unpleasant feelings, and nine items (6 to 15) assess decentering skills with unpleasant thoughts. Why this imbalance? (3) Item 6 assesses to what extent participants notice an unpleasant thought. If yes, a number of items explore about this unpleasant thought. And regardless of the response to item 6, other items explore the ability to distance (item 13), to think differently (item 14), to distract (item 15), not to feel sucked by such negative thought (item 12). I am confused about the interpretation of the responses to these items (12 to 15) in the case of responding "not at all" to item 6. In other words, if I do not notice any unpleasant thought, how could I report about my ability to decenter from an absent thought? But the application "forces" me to respond, otherwise I will lose money... (4) More about item 6. Is "noticing" being used as an index of presence of an unpleasant thought, or as an actual decentering skill? Within ACT, we state that it is not the same thinking than noticing a thought, the latter involving certain degree of distance or de-fusion. But the fact that other items are presented or not contingent upon the response to item 6, makes me wonder. (5) Lastly, why there is no item that assesses the presence of unpleasant feelings since the last beep, something like item 6 but with feelings? Same as before, without checking for the presence
--	--

	or not of an unpleasant feeling, responses to items 3 to 5 seem difficult to interpret. (6) Bonus question. What does “Branch 1, Branch 2, and Branch 5” mean (Table 1, last three rows). I do not recall reading about branches in the manuscript. 6. I miss more information about the psychometric properties (validity, reliability) of the adolescent versions of the assessment instruments employed. 7. I congratulate the authors for including measures of far transfer effects, they can yield very interesting and informative results. But I would appreciate a little rationale on the chosen variables, e.g., the evidence sustaining the transfer effects to affective cognitive control and body perspective, or the hypothesized mechanisms underlying such effects. In other words, why these variables and tasks, and not others? It would add coherence to the assessment. 8. Considering that this trial could be highly demanding for participants, I believe data analyses would benefit from controlling for the amount of engagement in the tasks and assessments. REVIEW CHECKLIST Abstract: Authors only include information about the primary outcome measures (decentering) and secondary mental health outcomes. However, to seize the scope of the present trial, I suggest the authors include information about the other measures that will be collected (emotion regulation and transfer effects). All in all, I congratulate the authors for a much-needed randomized controlled trial with adolescents. In my humble opinion, this is well-designed trial with a solid and coherent rationale. Thank you for letting me participate in this discussion, and hope my comments and suggestions will be helpful.
--	---

REVIEWER	Griffiths, ?Helen The University of Edinburgh
REVIEW RETURNED	20-Dec-2021

GENERAL COMMENTS	This is an extremely well specified protocol that addresses important contemporary issues using an innovative design that is clearly and comprehensively set out. I have a few very minor suggestions for revision that would further enhance the high quality of this protocol and otherwise would strongly recommend that this manuscript is accepted for publication. Pg 1 line 54 replace (intern) with (inter) Pg 2 lines 33-36 punctuation required for (e.g. unpleasant feelings thoughts memories) Pg 8 Strengths section – it is stated that this research seeks to develop a skill that may improve well-being across the spread of adolescent mental health, from suffering through to flourishing. However, on page 12 it is stated that the target population is one that is ‘at increased risk of depression’ and subsequently a cohort that is experiencing ‘elevated symptoms of depression’. Given that this specific project is targeting a group with increased risk of depression, I suggest rewording the strengths section to reflect this Pg 11 define what is meant by ‘emotional unfolding’ (or provide citation?)
--

	Pg 13 lines 6-8 Clarify how many participants will be in each arm. On Pg 13 it specifies 48 participants, elsewhere (e.g. abstract, pg 15 and other sections) it specifies 57 participants per group Pg 15 lines 54-56 give example for week 4 exercise Pg 21 Table 1 Item 9 refers to “Social things, Professional things, A combination of these, None of these” – ‘professional things’ would not be applicable to a group of adolescents recruited through schools. Is this the item that was intended?
--	--

VERSION 1 – AUTHOR RESPONSE

Reviewer: 1

Dr. Sonsoles Valdivia-Salas, University of Zaragoza

Comments to the Author:

GENERAL COMMENTS

Thank you for inviting me to review this protocol for a randomized controlled trial to assess the effect of decentering on adolescents’ emotion reactivity, emotion regulation skills, mental health, and cognitive and metacognitive abilities.

This a very welcome study, especially because of the target population: adolescents. As the authors point out in the Introduction, the evidence on the benefits of decentering in mental health is growing and consistent. But this evidence is limited to studies with adults.

Considering that mental health problems do not appear abruptly but develop over time, focusing on children and adolescents seems of vital importance for the prevention of more severe pathologies during adulthood. In addition, adolescence is, in itself, a stage of development in which profound changes occur that could affect their mental health. For these reasons, recruiting adolescents as the target population seems like a wise choice, and I congratulate the authors for this.

We thank the reviewer for their positive comments, enthusiasm for the study, and for their detailed feedback.

I have no objections to the protocol rationale, design, evaluation, and dissemination, but there are some issues I would like to discuss about.

1. In the Article Summary, Limitation’s subsection (p. 2), the authors mention the risk of non-adherence because of a lacking face-to-face contact and, possibly, because of the protocol being highly demanding. In any case, I would say these two aspects will inform of how much face-to-face contact and demand is necessary for adolescents to engage in decentering skill training. However, I missed mention to the possible contamination between experimental and control conditions in the remote case that participants get to know or meet and talk about the procedures. This could be the case if a number of adolescents were recruited from the same school. Do the authors have a plan to avoid possible contamination? Please state.

Our apologies that this was unclear. Contamination is possible due to participants being recruited from the same school, but is limited by two factors. Firstly, participants are briefed and debriefed individually, so are not aware of other participants in the trial. Furthermore, we ask participants at briefing to keep their group membership confidential. To reflect this, we have now added a line to the “Consent and Data Collection Procedures” section on page 22 which reads “To discourage possible contamination, participants will be asked to keep the content of the study confidential from others.”

2. In the Background section, the authors state: “Evidence also indicates self-reported decentering traits in adolescents are associated with fewer overall symptoms of anxiety and depression” (p. 4, lines 15-17) and cite four studies on the role of psychological in/flexibility on mental health symptoms. Hence, I wonder why a measure of psychological in/flexibility is not included in the assessment battery. This would allow for further investigation on the relation between constructs, or even the role of psychological in/flexibility (with proven effects on adolescents’ mental health and wellbeing) on the observed outcomes.

Our team and others have argued elsewhere that psychological flexibility is conceptually similar to the construct of decentering (Bernstein et al. 2015, Bennett et al. 2020). We also noted how measures of decentering (e.g. the experiences questionnaire) and psychological flexibility (e.g. the acceptance and action questionnaire) have been used side-by-side in different studies. So, we agree that it will be interesting to investigate the connection between these constructs (for example, is decentering just one part of psychological flexibility or is psychological flexibility better explained by decentering?). Unfortunately, there are limits to the number of questions we can answer in a single trial and we are focused on a relatively simpler matter – does decentering training work for adolescents, and if so what are the cognitive and mental health benefits? If we can confidently address these issues, then we believe we will be in a much stronger position to delineate decentering from other therapeutic processes in future research. Also, and on a pragmatic level, the battery of tasks and planned analyses is already quite demanding of participant/researcher time. We therefore doubt if there is capacity to include any additional measures at this time.

3. The N size was confusing. At different points through the Abstract and Methods section, it reads 48 (p.8, line 6), 57 (Abstract and p.9, line 54), and 54 (p.10, line 15) per group. Please clarify.

We apologise sincerely for this error and thank the reviewer for their attention to detail. The N size should now read 57 per group in all sections.

4. As stated in the Intervention subsection (p. 10), Week 5 of the decentering training is a revision week. How will this revision be conducted? Any new task or exercise? Something like a narrative summary of the skills trained during the previous weeks? Repetition of the exercises practiced before? More information will be appreciated for the sake of replication.

Thank you for your comment. Our final manuscript will include a more detailed description of the therapeutic materials, which we will also make freely available on OSF. For the current manuscript, we are working within a more restricted word count, hence the lack of week by week description of exact content of the intervention. We have now amended this sentence to read: “Week 5 is a revision week, during which participants will be encouraged to practice a different technique from weeks 1 to 4 each day.”

5. As part of the decentering outcome measures, two ESM decentering items will be presented to the participants (p. 13). These two items assess the self-reported ability to distance from unpleasant feelings and thoughts. These are items 4 and 13 of a total 16 included in Table 1. I am a little confused with this ESM:

The original manuscript split the discussion of ESM items by trial outcome. That is, there are primary decentering ESM items and secondary emotional reactivity ESM items. We still believe this is an appropriate way to structure the manuscript but admit it may have created confusion around the ESM items. We therefore included the following section when first introducing ESM items: “All ESM items are described in Table 1. Some items relate primarily to the use of decentering skills. Other ESM

items relate to secondary mental health outcomes and are described in the next section. However, all ESM items will be administered during each sample point.”

(1) I am assuming items 4 and 13 are not presented alone, but in combination with the whole ESM, is this right?

This is correct. Each ESM item listed in the table are presented one-at-a-time, in a single sitting. This includes items 4 and 13. We have explained this in the updated ESM section (described above).

(2) Table 1 lists a total of 16 items, most of them will be presented always, only a few will be presented contingent upon a particular response. Three items (3, 4, 5) assess decentering skills with unpleasant feelings, and nine items (6 to 15) assess decentering skills with unpleasant thoughts. Why this imbalance?

The imbalance as you describe is because separate questions are needed to estimate the onset/content of unpleasant feelings and unpleasant thoughts. Items 2i-2viii check whether participants experienced negative feelings. Items 7-10 check whether participants experienced some upsetting cognitive event. These are clearly different events requiring different questions. Otherwise, we believe that items are well-balanced. There are two questions on decentering skills with feelings (3 and 4), and there are two questions on decentering skills with thoughts (items 12 and 13). Distraction is an alternative regulation option that could be applied for both feelings (item 5) and thoughts (item 15). Reappraisal was included as an another broad regulation skill (item 14) that has been linked to decentering (Powers and LaBar, 2019) - the item is worded so not to explicitly refer to thoughts or feelings. We have updated Table 2 column 4 to clarify this.

(3) Item 6 assesses to what extent participants notice an unpleasant thought. If yes, a number of items explore about this unpleasant thought. And regardless of the response to item 6, other items explore the ability to distance (item 13), to think differently (item 14), to distract (item 15), not to feel sucked by such negative thought (item 12). I am confused about the interpretation of the responses to these items (12 to 15) in the case of responding “not at all” to item 6. In other words, if I do not notice any unpleasant thought, how could I report about my ability to decenter from an absent thought? But the application “forces” me to respond, otherwise I will lose money...

The decision to require a response is necessary for our planned ESM analysis of the lagged effect of decentering on affect. This is described in the section ‘ESM Emotional Reactivity’ - our plan is to model if later mood/feelings (item 1) is predicted by use of decentering skills with thoughts (item 13) controlling for the actual onset of a difficult thought (item 6). That is, if items 12-13 were excluded contingent on the participants answer to item 6, then decentering data points will be missing, non-randomly. And the inclusion of non-random missing data points will limit our ability to model lagged effects.

We appreciate the reviewers concerns around the meaningfulness of items 12-15 if item 6 is answered ‘not at all’. However, we believe that responses to 12-15 can still make sense. Participants simply indicated zero to questions 12-15 to indicate that they did not try to distract/think differently/distance. This was also our observation during piloting. However, and for clarity, we will discuss this into our ESM briefing for each participant.

(4) More about item 6. Is “noticing” being used as an index of presence of an unpleasant thought, or as an actual decentering skill? Within ACT, we state that it is not the same thinking than noticing a thought, the latter involving certain degree of distance or de-fusion. But the fact that other items are presented or not contingent upon the response to item 6, makes me wonder.

The purpose of item 6 is to estimate whether participants experienced a difficult thought from which they could decenter, immerse or distract. It will not be used as a measure of decentering itself. To indicate this, we have updated Table 1 – column 4 (outcome), to read “purpose”.

We understand and agree that, from a clinician’s perspective, ‘noticing’ and ‘thinking’ are different and that the former likely implies some degree of distance. That said, accurately answering a question on whether certain of ‘thinking’ occurs also implies a level of meta-cognitive monitoring. This speaks to the challenges in precisely measuring decentering and related meta-cognitive phenomenon. Tackling this challenge is beyond the scope of the current project. For now, we believe that using the term ‘noticing’ and a Likert scale is age-appropriate and gives us our ‘best-bet’ when estimating the onset of unpleasant thoughts and opportunities use the trained decentering skills. This is because the term ‘noticing’ can still be relatively independent of decentering – people can notice their difficult thoughts without necessarily applying therapeutic skills they have learned like temporal, spatial, hypothetical or objective distancing.

(5) Lastly, why there is no item that assesses the presence of unpleasant feelings since the last beep, something like item 6 but with feelings? Same as before, without checking for the presence or not of an unpleasant feeling, responses to items 3 to 5 seem difficult to interpret.

This is a limitation and this will be discussed in the final manuscript. Question 2 assess the presence of both unpleasant and pleasant feelings at each time point. The ESM items are already lengthy, and therefore, it was decided to leverage these items to assess presence of unpleasant feelings rather than include an additional item about the onset of feelings since the last beep. While not ideal we felt this was a fair compromise between the richness of data collected and participant effort.

(6) Bonus question. What does “Branch 1, Branch 2, and Branch 5” mean (Table 1, last three rows). I do not recall reading about branches in the manuscript.

‘Branch 5’ should read ‘Branch 3’. This has been updated in the manuscript. ‘Branch’ items are now described in the manuscript. If a participant says ‘not at all’ to item 6, then they complete ‘branch’ items 1-3’ in lieu of items 8-10. This guarantees that participants always complete the same number of items. The following section has now been included: “One item will assess the occurrence of unpleasant thoughts: ‘since the last beep, I noticed an unpleasant thought or memory’. This will allow us to control for the presence of unpleasant cognitive events from which participants can decenter. If the participant’s response is anything other than ‘not at all’, they will then complete three items to the content of these thought (Table 1, items 8-10). If the participant’s response is ‘not at all’, they will complete three unrelated ‘branching’ items (Table 1, branch items 1-3). This means the same number of item will always be administered.”

6. I miss more information about the psychometric properties (validity, reliability) of the adolescent versions of the assessment instruments employed.

Thank you for your comment. Details of validity and reliability have been added on pages 13, 16, 17, 18 for each of the measures.

7. I congratulate the authors for including measures of far transfer effects, they can yield very interesting and informative results. But I would appreciate a little rationale on the chosen variables, e.g., the evidence sustaining the transfer effects to affective cognitive control and body perspective, or the hypothesized mechanisms underlying such effects. In other words, why these variables and tasks, and not others? It would add coherence to the assessment.

Thank you for your comment. We agree that there are a vast array of far transfer effects which could have been included in the study. Due to the workload already placed on participants, we elected to

include tasks to address only our most important questions from a theoretically driven perspective. Powers & LaBar (2019) suggest a model of decentering comprised of three parts – self-projection, affective self-reflection, and cognitive control. Thus, we included an stop signal task to explore cognitive control, and further expanded on it by including an affective or emotional element. This will allow us to explore relationships between decentering and cognitive control in an affective context. Furthermore, we elected to explore self-projection and its relationship with self-rated decentering through an Own Body Task, in which participants must effortfully alter their self-perspective. We have added a clarification into page 20 the manuscript reading: These include (1) affective cognitive control and (2) self-perspective processing, both of which have been posited as theoretical components of decentering.

8. Considering that this trial could be highly demanding for participants, I believe data analyses would benefit from controlling for the amount of engagement in the tasks and assessments.

The following section on Data Quality has been included.

“Data quality

‘Checks of data quality will be incorporated during data collection and analysis. First, four ‘catch questions’ will be embedded in self-report assessments to assess whether participants actively read questions (e.g. ‘For this question, please select ‘Almost Always?’). Second, participants self-rate the quality of their data at the end of each collection period, e.g. ‘On the scale from 0 (not at all) to 100 (very much), how much did you properly engage with the questionnaires? Please answer honestly, you will still be paid regardless of your response.’ Piloting revealed this to be a useful approach. Third, all the standard markers of appropriate behavioural performance will be checked when analysing task data (e.g. reaction times shorter 250 ms and greater 3000 ms; chance-level accuracy).”

REVIEW CHECKLIST

Abstract: Authors only include information about the primary outcome measures (decentering) and secondary mental health outcomes. However, to seize the scope of the present trial, I suggest the authors include information about the other measures that will be collected (emotion regulation and transfer effects).

This is a very welcome suggestion, however due to the word limit on our abstract, we will not be able to put in additional information at this time.

All in all, I congratulate the authors for a much-needed randomized controlled trial with adolescents. In my humble opinion, this is well-designed trial with a solid and coherent rationale. Thank you for letting me participate in this discussion, and hope my comments and suggestions will be helpful.

Reviewer: 2

Dr Helen Griffiths, The University of Edinburgh

Comments to the Author:

This is an extremely well specified protocol that addresses important contemporary issues using an innovative design that is clearly and comprehensively set out. I have a few very minor suggestions for revision that would further enhance the high quality of this protocol and otherwise would strongly recommend that this manuscript is accepted for publication.

We thank the reviewer for their positive comments, enthusiasm for the study, and for their detailed feedback.

Pg 1 line 54 replace (intern) with (inter)

Thank you, we have amended the manuscript accordingly.

Pg 2 lines 33-36 punctuation required for (e.g. unpleasant feelings thoughts memories)

Thank you, we have amended the manuscript accordingly.

Pg 8 Strengths section – it is stated that this research seeks to develop a skill that may improve well-being across the spread of adolescent mental health, from suffering through to flourishing. However, on page 12 it is stated that the target population is one that is ‘at increased risk of depression’ and subsequently a cohort that is experiencing ‘elevated symptoms of depression’. Given that this specific project is targeting a group with increased risk of depression, I suggest rewording the strengths section to reflect this

Thank you for your comment. We have now reworded the “Strengths” section to read: “We aim to strengthen an active and pan-therapeutic skill that features across different psychological interventions, and may improve well-being in adolescents at high risk of mental health difficulty.”

Pg 11 define what is meant by ‘emotional unfolding’ (or provide citation?)

The term ‘emotional unfolding’ relates to an earlier draft of our protocol. This has been updated to ‘emotional reactivity’ which more accurately reflects our secondary analysis plan as described in the manuscript.

Pg 13 lines 6-8 Clarify how many participants will be in each arm. On Pg 13 it specifies 48 participants, elsewhere (e.g. abstract, pg 15 and other sections) it specifies 57 participants per group

We apologise sincerely for this error and thank the reviewer for their attention to detail. The N size should now read 57 per group in all sections.

Pg 15 lines 54-56 give example for week 4 exercise

We have now added the following example to the manuscript: (e.g. ‘what is the effect of reliving a difficult memory from the perspective of an objective observer?’)

Pg 21 Table 1 Item 9 refers to “Social things, Professional things, A combination of these, None of these” – ‘professional things’ would not be applicable to a group of adolescents recruited through schools. Is this the item that was intended?

Our youth advisors regularly describe how academic performance, college and career ambitions as well as any part-time employment can be a source of stress. Understandably, this can be especially true of older adolescents who are recruited in our study. ‘Professional things’ is therefore used to generally estimate these sorts of stressors and this will be described in our briefing. This question is also included, not as a primary outcome, but instead to provide some basic insight into the context of adolescent difficulties.

Reviewer: 1

Competing interests of Reviewer: I declare I have not competing interests

Reviewer: 2

Competing interests of Reviewer: None

VERSION 2 – REVIEW

REVIEWER	Valdivia-Salas, Sonsoles University of Zaragoza
REVIEW RETURNED	09-Feb-2022

GENERAL COMMENTS	It has been a pleasure to read both the answers to the reviewers' comments and the revised version of the manuscript. The authors have addressed carefully all the issues I raised in my review and amended the manuscript appropriately when necessary. Congratulations on a fine work.
--

REVIEWER	Griffiths, ?Helen The University of Edinburgh
REVIEW RETURNED	26-Jan-2022

GENERAL COMMENTS	The minor revisions to the manuscript answer the reviewers' earlier comments and improve on a protocol that was already of high quality. I recommend the manuscript for publication.
--